# Transferring Grains from Single-Grain Luminescence Discs to SEM Specimen Stubs

**DOI:** 10.3390/mps2040087

**Published:** 2019-11-21

**Authors:** Isa Doverbratt, Helena Alexanderson

**Affiliations:** 1Department of Geology, Lund University, Sölvegatan 12, SE-223 62 Lund, Sweden; isa.doverbratt@geol.lu.se; 2Department of Geosciences, UiT the Arctic University of Norway, PB 6050 Langnes, N-9037 Tromsø, Norway

**Keywords:** luminescence, single-grain, SEM, sample holders

## Abstract

The grain transfer protocol presents a step-by-step guide on how to successfully transfer positioned grains from a single-grain luminescence disc to a scanning electron microscope (SEM) specimen stub and how to transport them between laboratories. Single-grain luminescence analysis allows the determination of luminescence characteristics for individual sand-sized grains. By combining such luminescence data with other grain properties such as geochemical composition, shape, or structure also at single-grain level, it is possible to investigate factors controlling luminescence signals or study other material properties. The non-luminescence properties are typically measured in another instrument; thus, grains need to be transferred between machines and sample holders, and sometimes also between laboratories. It is then important that the position of each grain is known and stable so that the properties from the same grain are compared. By providing an easily observable orientation marker on the specimen stub, the hundred numbered grains from the single-grain disc can be transferred and later identified when analyzed in the SEM.

## 1. Introduction

With the development of single-grain luminescence analysis [1,2], it has become possible to measure the luminescence signals of individual quartz or feldspar grains. From a dating perspective, this has provided valuable information on dose distributions that can be used to evaluate, for example, sediment mixtures or incomplete bleaching [3,4], information that is partly lost in multigrain aliquots because of the averaging of grains that takes place [5]. Single-grain luminescence has also revealed the variability in luminescence characteristics between grains within a single sample [6]. While this may complicate age determination [7,8], it also offers possibilities to study and compare luminescence characteristics on the level of individual grains, with the aim of understanding causes and controls of luminescence of quartz or feldspar. However, this may require the additional analysis of grain properties that cannot be carried out in a single-grain luminescence reader, for example of geochemical composition, crystal structure, or grain shape. To do such analyses, grains need to be transferred from the luminescence reader sample holders (single-grain discs) to the sample holders of another instrument, while keeping track of which grain is which to be able to compare properties on the grain level. Potential applications of such comparative analyses include determining the K content in feldspar grains, identifying impurities in quartz or feldspar grains, studying the effect of hydrofluoric acid or other chemical treatments, evaluating bleaching for different transport pathways by analyzing grain shape and surface textures (of nonetched grains), and comparing luminescence signals or doses at the grain level. In some cases, it may also be interesting to simply review the grains in a scanning electron microscope (SEM) or optical microscope to determine grain integrity (missing grains, multiple grains per hole, grain appearance, etc.). In this paper, we present a protocol that allows for successful transfer of grains from single-grain discs to SEM specimen stubs.

## 2. Experimental Design

The main focus in this procedure is how to transfer grains from a single-grain disc (SG disc) to an SEM stub and transport the analyzed single grains, intact and in order, from one laboratory to another. The most crucial part in this procedure is sample orientation in order to identify grains, and noting that the grains on the SEM specimen stub will be the mirror image of the grains on the SG disc.

Four samples previously dated at the Lund Luminescence Laboratory, Lund University, Sweden, were used in this study. The grains were transferred from single-grain discs to SEM specimen stubs in the luminescence laboratory at the University of Sheffield, UK, and then transported by train, plane, and bus to Lund University, Sweden. Apart from the sample holders, the protocol requires few special materials and can be applied in most laboratories and with various transport distances.

### 2.1. Materials

The grains used for the transfer were of quartz in the 180–250 µm grain size fraction. The grains were extracted from Late Quaternary sediments from four sites in Sweden and Norway (Table 1) according to procedures described in [9].

### 2.2. Equipment

Instruments◦Risø TL/OSL reader model DA-15 with a single-grain attachment [1] at the luminescence laboratory at the Department of Geography, University of Sheffield, United Kingdom◦Tescan Mira3 High Resolution Schottky FE-SEM equipped with Oxford EDS at the Department of Geology, Lund University, Sweden◦Stereo Microscope, Zeiss Stemi 2000-C, equipped with a camera, AxioCam ERc 5s, at the Department of Geology, Lund University, SwedenSample holders◦Single-grain aluminum discs, 9.7 mm in diameter and 1 mm thick, with one hundred 300 µm holes in a 10 × 10 grid◦Specimen mount (stub), aluminum, 1/2” slotted head, 1/8” pinOther◦Carbon adhesive tabs, 12 mm diameter, Electron Microscopy Sciences◦Scissors◦Small container, plastic or other material, about 0.5–1 cm higher than the stub height◦Styrofoam, thickness about 1 cm (corresponding to height of stub rods)◦Tweezers

## 3. Procedure

### 3.1. Preparing the Transportation Container

Find a container of appropriate size with a lid. An appropriate size is one that suits the number of samples and allows a safe transportation of the samples, that is keeping the samples upright.Cut a piece of Styrofoam into the desired shape to fit the bottom of the container (Figure 1). Any other flexible but firm material may also be used.Prepare holes in the Styrofoam to fit the SEM specimen stubs (Figure 1). Hold the stubs upright (broad side up) and push down the rods into the Styrofoam. If necessary, use a sharp object (for example, a pen) to pierce the holes for the stub rods.Mark each hole with a number, for example 1–6 (Figure 1).

### 3.2. Preparing the SEM Specimen Stubs

Use scissors to cut the carbon adhesive tabs so that there is one straight side to the previously circular piece.Attach the precut tab centered on an SEM specimen stub, and leave the transparent plastic film on (Figure 2).Place the stub in the container (Section 3.1) with the rod in one of the holes and the tape side up.

### 3.3. Transferring Grains from the Single-Grain Disc to the SEM Specimen Stub

After OSL measurements, take the wheel out of the reader and place it on a stable and horizontal surface.Take one of the pre-prepared SEM specimen stubs out of the container and remove the transparent plastic film with tweezers.Remember to note the orientation of the holes in the disc and the straight side of the carbon tape.Hold the stub on its rod and carefully attach the sticky carbon tape to the SG disc, with the disc still in the wheel. Make sure that the stub is centered and placed with the cut tape side at the “no-hole side” of the SG disc (Figure 2). This enables you to keep track of the orientation, that is to be able to use the numbered grain positions from the reader software. Attention: The grain numbering on the carbon tape is the mirror image of the grains on the SG disc (Figure 3). The grains will also be upside-down compared to the SG disc.When the SEM specimen stub is fitted to the SG disc, turn it into an upright position and tap the disc gently with the tweezers.Remove the SG disc from the carbon tape using tweezers or your fingernails. You will now be able to see the grains on the carbon tape.Place the stub carefully in the container prepared with Styrofoam. The rod should go into one of the rod holes, and the tape side should face upwards.Make a note of which sample is in which hole (Figure 4).Put the lid on the container, and, if necessary, tape it to keep it in place.

### 3.4. Transport

During transport, make sure that the container stays upright and is packaged securely to ensure minimal movement of sample holders.

#### Additional Information

To orient the grid of grains once the sample is in the SEM, it is advisable to take an overview picture of the stub. The cut side of the carbon tape is clearly seen in the SEM (Figure 5), which allows the grains to be numbered (1–100). Pictures of each disc may also be taken with a stereo microscope equipped with a camera prior to SEM analysis.

## 4. Results

The success of the transfer was evaluated by (1) counting the number of empty holes in the single-grain discs by inspecting the OSL signals from each position; (2) counting the number of positions in the 10 × 10 grid on the SEM stubs that are filled with grain(s), as determined from an image taken in the SEM (Figure 5); and (3) comparing the numbers. On average, grains at 94% ± 6% of the positions were successfully transferred from the disc to the stub (Table 2). Some of the missing grains can, however, still be seen on the stub, but are not located in a grid position (Figure 5), either because they moved slightly within the grid or they ended up off the grid.

Even samples with multiple grains at almost all positions, such as the easily shattered quartz of sample 15012, retained the grid pattern (Figure 5b) and made it possible to identify position, though in some cases the piles of grains at neighboring positions merged. It is, therefore, fairly straightforward to identify grains that have moved, and we recommend that grains with a position off the grid are excluded from further analysis, since their connection to a specific single-grain luminescence measurement would be uncertain.

Concerns that may arise during SEM analysis of grains transferred according to the protocol presented here include the identification of individual grains, the orientation of grains, and contamination from the SG disc. Regarding the identification of individual grains, we found that having an overview image where (at least) grains in grid corners have been numbered (compare Figure 5) greatly facilitates finding specific grains based on their location and appearance, also at high magnification. 

The second concern, that the same surface is not exposed for imaging or analysis before and after transfer (because of the grains being upside-down), may or may not be an issue depending on what type of analysis is carried out. Some analyses, including luminescence, retrieve information also from the interior of grains, and for these it should be less of a concern. This is something that could be studied further, as could the risk of contamination from the SG disc to the grains. If the discs are visually inspected in a microscope prior to use, and damaged or dirty discs are discarded, we consider the risk to be negligible, but we have not studied this specifically. Any such contamination would also be surficial and would likely not significantly influence measurements that include deeper parts of the grains, or whole grains.

## Figures and Tables

**Figure 1 mps-02-00087-f001:**
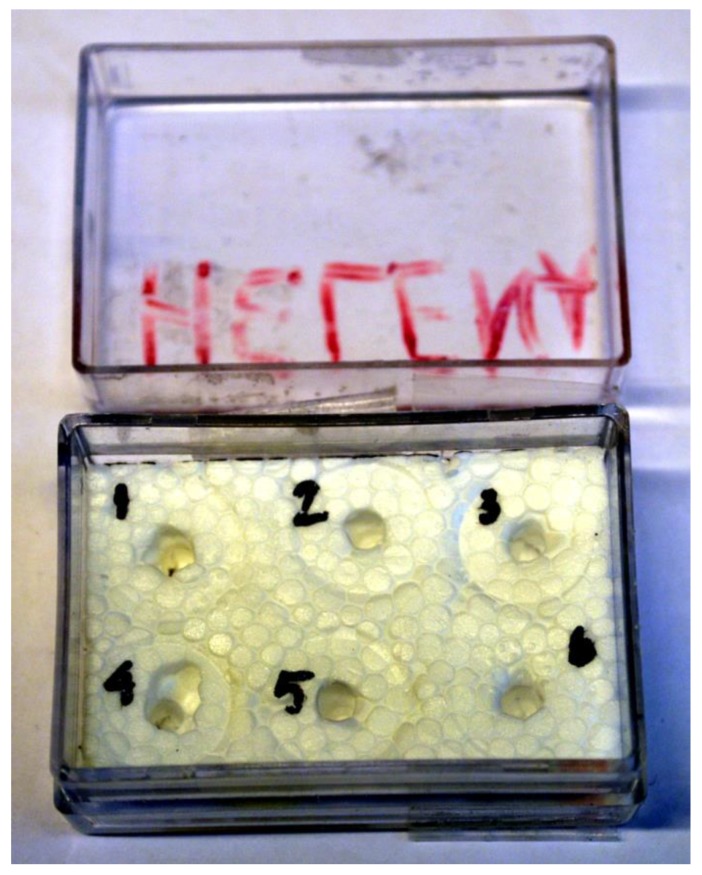
Container (3.5 × 5 cm) with lid, prepared with Styrofoam and holes for the scanning electron microscope (SEM) specimen stubs. This specific container can hold up to six stubs.

**Figure 2 mps-02-00087-f002:**
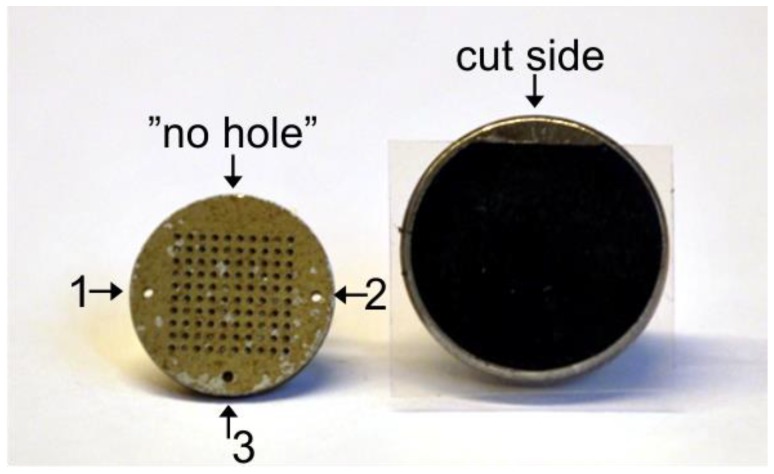
To the left is a single-grain disc with the “no-hole” side of the 10 × 10 grid in the uppermost part of the disc. Numbers 1–3 indicate the orientation of the holes, as registered in the luminescence reader software (SequenceEditor). To the right is an SEM specimen stub prepared with black, cut carbon adhesive tab and the transparent plastic film left on.

**Figure 3 mps-02-00087-f003:**
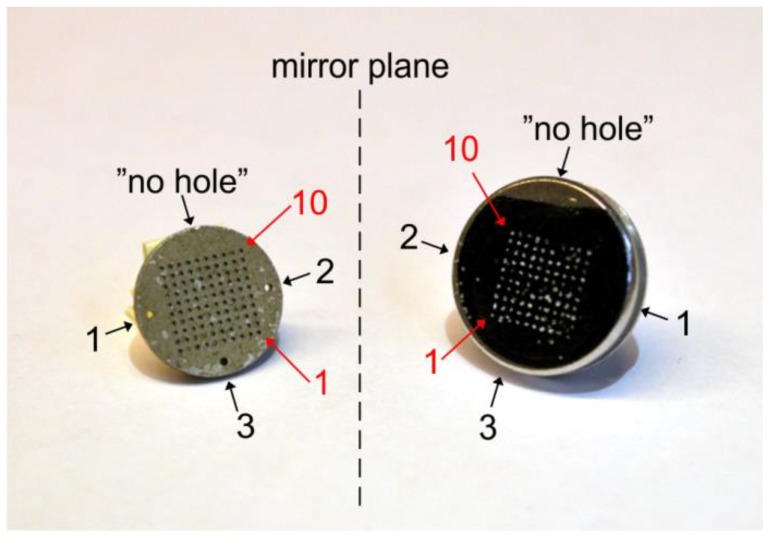
An SG disc (**left**) after the grains have been transferred to the carbon tape on the SEM specimen stub (**right**). The position of the grains on the stub is the mirror image of how the grains were positioned on the disc. Numbers 1–3 (in black) highlight the orientation of the holes, and numbers 1 and 10 (in red) highlight the grain positions on the disc as recorded in the OSL data file by the reader software.

**Figure 4 mps-02-00087-f004:**
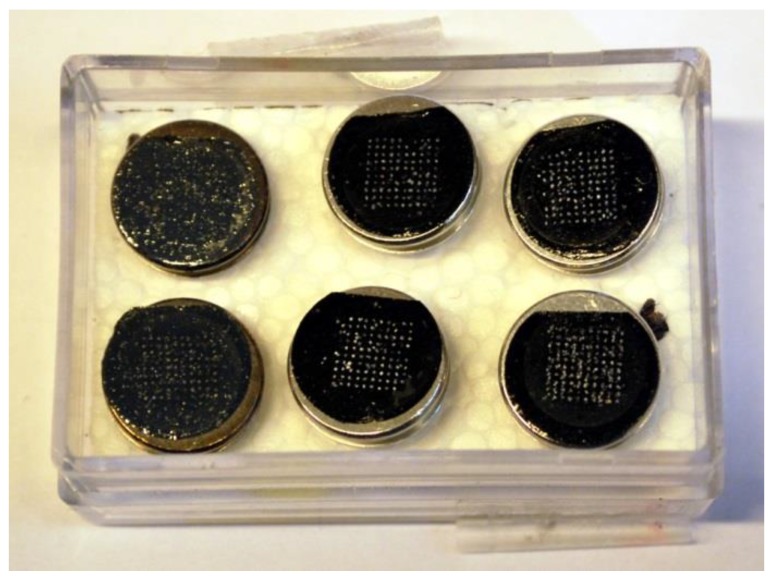
Six SEM specimen stubs in the container prepared with Styrofoam (the two stubs to the left in this figure have a different appearance due to a Pd/Pt coating prior to SEM analysis).

**Figure 5 mps-02-00087-f005:**
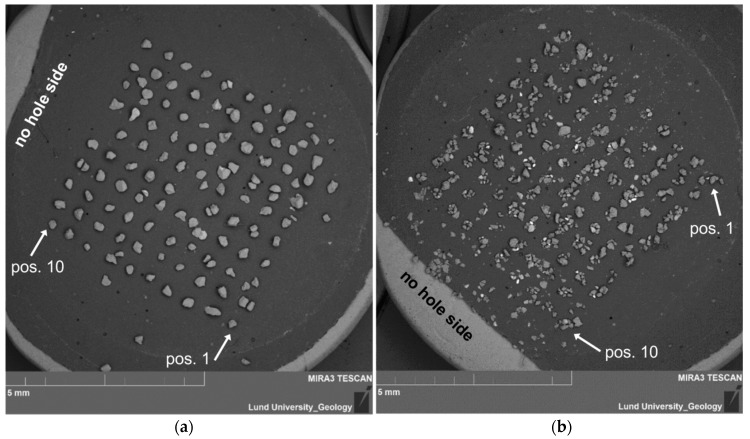
Examples of SEM images of transferred grains on stubs. The 10 × 10 grid can be seen clearly both where there is (**a**) mainly one grain per position (disc 2, sample Lund-15064) and (**b**) multiple grains per position (disc 6, sample Lund-15012).

**Table 1 mps-02-00087-t001:** Sample information.

Lab. No Lund-	Site	Sediment	Disc	Reference
12057	Rauvospakka, N Sweden	glacilacustrine silty sand	4, 5	[10]
13031	Orsa, C Sweden	aeolian sand	1, 7, 8, 9	[11]
15012	Skorgenes, W Norway	glacifluvial sand	6	[12]
15064	Skogalund, SW Sweden	aeolian sand	2, 3	[9]

**Table 2 mps-02-00087-t002:** Number of positions on the discs and stubs, respectively, that are filled with grains.

Disc	#Filled Positions on Disc	#Filled Positions on Stub	Transfer Success (%)
1	100	90	90.0
2	97	98 ^1^	101.0 ^1^
3	100	97	97.0
4	99	90	90.9
5	100	93	93.0
6	100	100 ^2^	100.0 ^2^
7	98	96	98.0
8	100	95	95.0
9	100	83 ^3^	83 ^3^

^1^ There were no OSL signals from three SG positions, but observed in the SEM, only two positions are empty; ^2^ The transfer of sample 15012 from the disc to the stub was the most successful. However, the quartz was quite brittle and shattered easily. Therefore, there are several smaller grains in each position; ^3^ This is the least successful transfer during which grains have been moved around, and many grains do not logically fit in a specific position.

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
