# Peer review of "Transferring Grains from Single-Grain Luminescence Discs to SEM Specimen Stubs"

_mps, 2019, doi:10.3390/mps2040087_

Round 1

Reviewer 1 Report

This manuscript has presented a simple but still important way of transferring a large amount of grains at the same time. It is suitable for a publication in the "Methods in Dating and Other Applications using Luminescence" topic. I suggest an acceptation. 

Author Response

Thank you.

Reviewer 2 Report

The authors discussed a protocol to transfer grains from luminescence discs to SEM stubs. Overall, the script is well written and deserves publication.

The following are some friendly remarks on English usage:

Page 1, line 12: "This" protocol: try to avoid the standalone "this" and define the protocol. Page 1, line 43: scanning electron microscope (SEM) specimen stubs; page 2, lines 53 and 54:scanning electron microscopy (SEM) specimen stubs. Consistence usage is required, and SEM is already defined. Page 2, line 77: in general "ca." is used to describe time, better use "about 0.5-1 cm" Page 2, line 82: "Find a container of appropriate size with lid" is better... Page 2, line 83, the use of "you" and "your" is a bit troubling: try to write in present simple passive... In general, unnecessary punctuations are road-blocks: try to use "for example" and "that is" instead of "e.g." and "i.e.", respectively. "an SEM"? or "a SEM"?: sound matters, right?. "train, plane and bus" Oxford comma?: "train, plane, and bus"

Author Response

All suggested language changes to the manuscript have been made. Please see below and tracked changes in the manuscript file.

Page 1, line 12: "This" protocol: try to avoid the standalone "this" and define the protocol. REWRITTEN

Page 1, line 43: scanning electron microscope (SEM) specimen stubs; page 2, lines 53 and 54:scanning electron microscopy (SEM) specimen stubs. Consistence usage is required, and SEM is already defined. UNECESSARY EXPLANATION DELETED

Page 2, line 77: in general "ca." is used to describe time, better use "about 0.5-1 cm" CHANGED

Page 2, line 82: "Find a container of appropriate size with lid" is better...CHANGED

Page 2, line 83, the use of "you" and "your" is a bit troubling: try to write in present simple passive...REWRITTEN

In general, unnecessary punctuations are road-blocks: try to use

"for example" and "that is" instead of "e.g." and "i.e.", respectively.REWRITTEN

"an SEM"? or "a SEM"?: sound matters, right?.CORRECTED

"train, plane and bus" Oxford comma?: "train, plane, and bus COMMA ADDED

Reviewer 3 Report

This may be a useful method for transferring grain samples for analysis with different instruments at different locations. However, the procedure has serious flaws. First, as indicated in the manuscript, some grains may not be successfully transferred from the SG disc to the stub and still be left on the SG disc or missing. Second, the grains appears to be loosely attached to the SG disc originally; and they may be missing or misplaced at a different locations on the stub during the transfer, causing difficulties in identifying each individual grains. And third, different sides or surfaces of the grains are exposed for analysis or imaging before and after the transfer. In addition, the side or surface of a grain that originally attached to the SG disc may be contaminated, thus not suitable for further analysis.

In fact, the original SG disc may be directly packed, shipped and analyzed with different instruments at different locations.

Author Response

“First, as indicated in the manuscript, some grains may not be successfully transferred from the SG disc to the stub and still be left on the SG disc or missing.”

Response: Yes, we do write in the manuscript that not all grains may be transferred so in that aspect the method is not a 100% success. But, the vast majority of grains are successfully transferred (average 94%, least successful 83% (Table 2 in manuscript)) and for most purposes this is quite sufficient. We therefore do not judge this as a serious flaw.

“Second, the grains appears to be loosely attached to the SG disc originally; and they may be missing or misplaced at a different locations on the stub during the transfer, causing difficulties in identifying each individual grains.”

Response: The grains are not ‘glued’ to the disc by silicon spray or similar but are only placed in the holes of the SG disc, and are therefore ‘loose’, though some grains may get stuck in the holes due to their size or shape. That the grains are loose rather makes the transfer to the stub easier, as it otherwise would have been more difficult to get the grains off the disc, which would have led to a higher risk of displacement (e.g. due to the vibration, picking or similar required to get the grains off). As mentioned in the manuscript, some grains may not be transferred to the stub and some grains may be displaced. However, since the 10x10 grid pattern is preserved, both missing and displaced grains can be identified as empty spaces in the grid or as grains lying outside the grid points. We consider it unlikely that grains have shifted position in such a way that they have ended up exactly at another grid point, where there was no other grain. This means that it is possible to recognize if there is a problem with missing or displaced grains, both on the level of a few grains and in rare cases for a whole disc (if the grid pattern should be only partly or poorly preserved), and any problematic grains/discs can be excluded from further analysis. From our experience, missing or displaced grains are a small problem (see response to first comment) and we do not consider this be a serious methodological flaw.

But we do recognize the problem and have added the following sentence in the manuscript (section 4): “It is therefore fairly straightforward to identify grains that have moved and we recommend that grains that have a position off the grid are excluded from further analysis, since their connection to a specific single-grain luminescence measurement would be uncertain.  “.

“And third, different sides or surfaces of the grains are exposed for analysis or imaging before and after the transfer. In addition, the side or surface of a grain that originally attached to the SG disc may be contaminated, thus not suitable for further analysis.”

Response: This is true. The grains would be upside-down on the stub compared to the disc, with the exception for any small movement (turning over) that may have happened during transfer, and this means that different parts of the grains would be exposed for the different measurements. Depending on which analyses will be done in the SEM (or other instrument) it may or may not make a difference which side is up or down. For analyses that take the whole grain or a large part of it into account it is less important, for surficial spot analyses it may be significant.

Regarding possible contamination: single-grain discs were inspected in an optic microscope prior to use, to check for any foreign material in any of the holes as well as the condition of the disc itself. Any scruffy discs were discarded while non-clean discs were cleaned. We are therefore confident that we do not have any significant contamination of visible non-grain material. ‘Invisible’ contamination may still occur though, and the likeliest source would then be the disc itself (i.e. streaks or flakes of aluminium). Such contamination could potentially be identified in the SEM (visually or analytically) but not easily. We have not tested for the risk of contamination from the disc to the grains, nor have we seen any other studies that have done this, either from luminescence measurements perspective or for other purposes – but would be happy to know of any such studies. However, depending on type of analysis, surficial contamination may or may not be a problem, and we have assumed that overall the risk is small.

We have added a note about the orientation in the manuscript (section 3.3): “The grains will also be upside-down compared to the SG disc.” and commented on this and the contamination issue in section 4: “The second concern, that the same surface is not exposed for imaging or analysis before and after transfer (due to the grains being upside-down), may or may not be an issue, depending on what type of analysis is carried out. Some analyses, including luminescence, retrieve information also from the interior of grains and for these it should be less of a concern. This is something that could be studied further, as could the risk of contamination from the SG disc to the grains. If the discs are visually inspected in a microscope prior to use and damaged or dirty discs are discarded, we consider the risk to be negligible, but we have not studied this specifically. Any such contamination would also be surficial and would likely not significantly influence measurements that include deeper parts of the grains, or whole grains.”

“In fact, the original SG disc may be directly packed, shipped and analyzed with different instruments at different locations.”

Response: As the reviewer also points out, “the grains appears to be loosely attached to the SG disc originally” and this means that even small vibrations or other shaking will cause grains to ‘jump’ out of the holes. Unless the holes on the disc are sealed, for example with tape, which can of course be done, they cannot be transported far without losing the grains. But even if the transport can be made with sealed discs, the discs may not fit directly into the other instrument and the grains still have to be transferred to another type of sample holder. It would be great if the grains, while still in the original disc, could be analysed in multiple ways, but at least for the conditions and equipment used in this study, this was not technically or practically possible.

Round 2

Reviewer 3 Report

I trust the authors have responded to my questions satisfactorily, therefore, agree to accept the improved manuscript for publication.